

# Bloch oscillations and the lack of the decay of the false vacuum in a one-dimensional quantum spin chain

O. Pomponio[1], M. A. Werner[2,4] G. Zaránd[2,4] and G. Takács[3,4]

**1** Dipartimento di Fisica e Astronomia dell'Università di Bologna and
Istituto Nazionale di Fisica Nucleare, Sezione di Bologna, Via Irnerio 46, 40126 Bologna, Italy
**2** BME-MTA Exotic Quantum Phases 'Lendület' Research Group,
Department of Theoretical Physics, Budapest University of Technology and Economics,
1111 Budapest, Budafoki út 8, Hungary
**3** BME-MTA Statistical Field Theory 'Lendület' Research Group,
Department of Theoretical Physics, Budapest University of Technology and Economics,
1111 Budapest, Budafoki út 8, Hungary
**4** MTA-BME Quantum Dynamics and Correlations Group, Eötvös Loránd Research Network

## Abstract

We consider the decay of the false vacuum, realised within a quantum quench into an anti-confining regime of the Ising spin chain with a magnetic field opposite to the initial magnetisation. Although the effective linear potential between the domain walls is repulsive, the time evolution of correlations still shows a suppression of the light cone and a reduction of vacuum decay via suppression of the growth of nucleated bubbles. The suppression of the bubble growth is a lattice effect, and can be assigned to emergent Bloch oscillations.



# 1 Introduction

The decay of the false vacuum is a famous scenario proposed in 1977 by Sidney Coleman to describe the dynamics of phase transitions in quantum field theory [1, 2], which plays an important role in particle physics and cosmology. In such a situation a system stuck in a metastable ('false') vacuum state transitions to the 'true' vacuum state by bubble nucleation and subsequent growth of the bubbles driven by the energy difference between the false and the true vacua. This is a non-equilibrium process, in which the expansion of the bubbles rapidly accelerates to the maximum possible velocity (the speed of light), and thus the true vacuum ultimately replaces the original false vacuum everywhere in space. Quantum bubble formation is, of course, not only relevant in cosmology, but it is the primary mechanism behind first order classical and quantum phase transitions and hysteresis.

Quantum quenches, i.e., sudden changes in a system's Hamiltonian [3,4] provide a paradigmatic protocol to study non-equilibrium quantum dynamics and relaxation in by now routinely engineered closed quantum systems [5–12], and thus provide a natural environment to test Coleman's scenario in various quantum systems. In fact, in global, translationally invariant quantum quenches, the initial state has a finite uniform energy density with respect to the post-quench Hamiltonian, and the system is therefore in a highly excited state. This highly excited configuration acts as a *source* of quasi-particle excitations, which may collide and thermalize with time. Indeed, within a semi-classical picture [3], these quasi-particle excitations drive equilibration by spreading correlation and entanglement across the system. In a large class of systems, including various spin chains with short-range interactions, or interacting bosonic or fermionic systems, quasi-particles have an upper bound on their velocity [13], leading typically to a distinctive light-cone pattern in time dependent correlation functions [3, 4, 14–19]. Preparing a system in the false vacuum as an initial state, therefore allows one to test quantum bubble nucleation in the laboratory [20–24].

A primary candidate to perform this experiment is the quantum Ising spin chain, governed by the Hamiltonian

$$H_{\text{QISC}} = -J \sum_i \left( \sigma_i^z \sigma_{i+1}^z + g \sigma_i^x + h \sigma_i^z \right) , \tag{1.1}$$

where $g$ and $h$ denote the transverse and the longitudinal magnetic fields, respectively. Here we consider the model in the thermodynamic limit, i.e., for a chain of infinite length, and set $J = 1$ which entails a choice of energy and time units. This model provides a paradigmatic example of quantum phase transition [25,26], and has also attracted interest as a model system for weak thermalization [27,28], and has been studied in the context of many-body scars [29]. In the ferromagnetic phase, $g < 1$, quasi-particles are domain walls, and the chain's dynamics may be understood in terms of quantum bubble nucleation [cf. Ref. [30] for a recent study in a different setting].

However, light-cone spreading of correlations is not completely generic [31]: confining forces, as a remarkable exception, can suppress the light-cone spreading of correlations. The prediction of *dynamical confinement* has indeed been confirmed recently in numerous systems and settings exhibiting confinement [32–40]. For the model (1.1) dynamical confinement occurs when initializing the system in its $g_0 < 1$ positive (spontaneous) magnetisation ground state with $m = \langle \sigma_i^z \rangle > 0$ and $h_0 = 0$, and then quenching to a Hamiltonian with some $g < 1$ and $h > 0$. In this case, the quench gives rise to oppositely moving domain walls (kink-antikink pairs), with a bubble of false vacuum with negative magnetisation $-m$ stretched between them (see Fig. 1.1a), which costs a potential energy proportional to the distance between the domain walls. The resulting confining force [41] inhibits the propagation of the domain walls to large distances, and prevents thermalization of the system within all time scales accessible to numerical simulations.

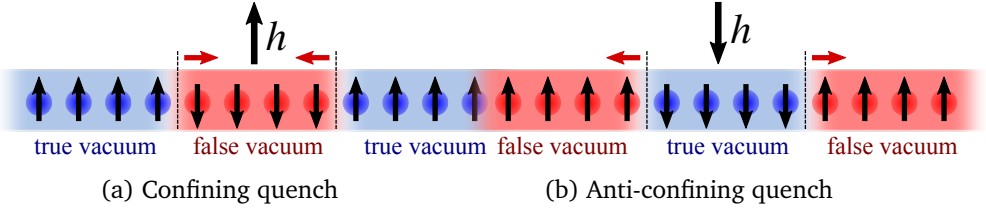

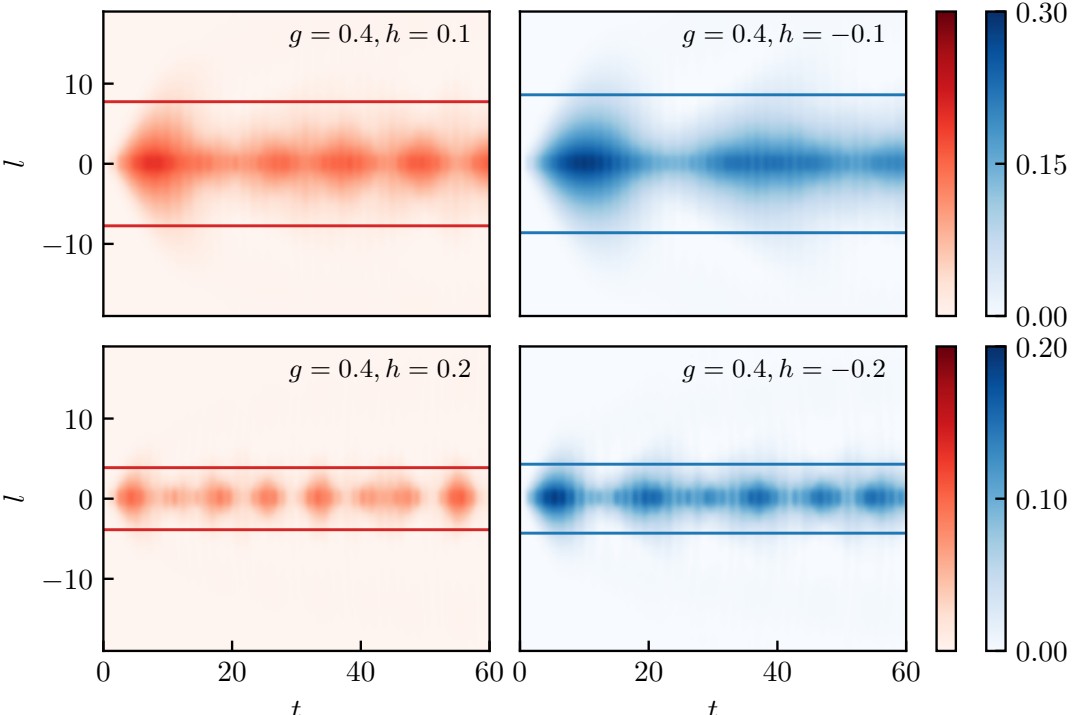

Figure 1.1: Upper panel: (a) For $h$ parallel to the initial magnetisation, bubbles nucleated during the quench contain the false vacuum. The corresponding attractive forces (red arrows) confine domain walls into 'mesons'. (b) For $h$ opposite to the initial magnetisation, nucleating bubbles contain the true vacuum. The induced repulsive forces (red arrows) accelerate the domain walls. Lower panel: Time evolution of the connected spin-spin correlation function $C_z(l, t) \equiv \left\langle \sigma_0^z(t) \sigma_l^z(t) \right\rangle_c$ for $g = 0.4$ in the confining regime, $h > 0$ (left), and in the anti-confining regime, $h < 0$ (right). Red vs. blue lines show average bubble sizes estimated using Eqs. (2.15) and (2.14), respectively.

It is intriguing to investigate what happens if we switch on a field in the opposite direction, $h < 0$, thereby initializing the system in the false vacuum of the final Hamiltonian. In this case, according to Coleman's scenario, nucleation must lead to bubbles of the true vacuum appearing inside a sea of false vacuum. The external field then *promotes* the expansion of bubbles, and generates a repulsive force between domain walls forming the bubbles, as illustrated in Fig. 1.1b. As a result, in the standard scenario the nucleated bubbles extend to the whole system and the system rapidly relaxes to an equilibrium or steady state around the true vacuum. In the quantum quench framework, this corresponds to the light-cone spreading of correlations, as predicted by the quasi-particle picture.

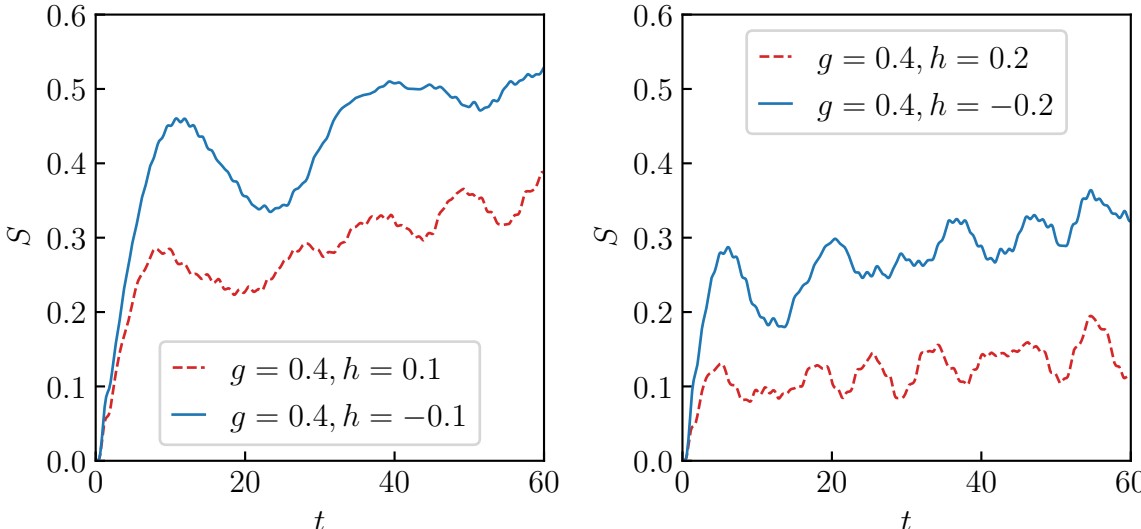

Figure 1.2: Time-dependence of entanglement entropy for the quenches displayed in Fig. 1.1. Note that anti-confining quenches (blue continuous lines) show a suppression of entropy growth similar to the confining case (red dashed lines), albeit with a larger magnitude of entanglement entropy generated during the quench.

Here we show by detailed simulations, however, that – quite counter-intuitively – the above scenario is violated in the Ising spin chain. The dynamics of spin-spin correlation function shown in Fig. 1.1 clearly indicates that the true vacuum bubbles do not expand indefinitely. In addition, computing the entanglement entropy between two halves of the system reveals that its initial linear growth is suppressed after a transitional period as shown in Fig. 1.2. For the confining quenches this effect was discovered in [31] and is explained by the suppression of light-cone spreading of correlations due to localisation of the quasi-particles by the confining force; however, in the anti-confining case this is unexpected in light of the force being repulsive.

We find that the relevant mechanism responsible for the lack of light-cone spreading of bubbles is not dynamical confinement: rather, the repulsive force leads to an oscillatory motion of the domain walls, known as Bloch oscillations. It was recently found that these oscillations slow down equilibration by preventing the fast propagation of excitations in other non-equilibrium settings, corresponding to starting the dynamics from a bipartite domain wall state [34], or from a random distribution of kinks [38]. Our findings are also consistent with recent numerical studies of the order parameter statistics in the quantum Ising spin chain [42].

Today, strongly correlated quantum many-body systems – including the quantum Ising spin chain itself [43] – can be routinely realised using ultra-cold atomic quantum simulators [44–48], or in a digital quantum computers [49]. These advances have triggered recently considerable interest in simulating the decay of the false vacuum [20–24]. Our results, demonstrating the suppression of false vacuum decay and providing directly observable signatures of the underlying Bloch oscillations, may be particularly interesting in this context.

## 2 Dynamics in the anti-confining regime

### 2.1 Quench setup and subsequent time evolution

As discussed in the introduction, we consider quantum quenches in the ferromagnetic phase of the quantum Ising spin chain (1.1). For simplicity, we start the quench from a fully aligned, positively polarised state, i.e. from the ground state with $g_0 = 0$ and $h_0 = 0$. We then quench to a finite transverse field, $g < 1$, and an "anti-confining" longitudinal field $h < 0$. Turning on a finite $g_0 \to g > 0$ creates a gas of domain wall excitations, which then move in the presence of the field $h < 0$.

The time evolution is numerically simulated using the infinite volume time evolving block decimation (iTEBD) method [50], which we use to compute the time evolution of the connected two-point equal time spin correlation function

$$
\begin{aligned}
C_z(l, t) &\equiv \left\langle \sigma_0^z(t) \sigma_l^z(t) \right\rangle_c \\
&= \left\langle \sigma_0^z(t) \sigma_l^z(t) \right\rangle - \left\langle \sigma_0^z(t) \right\rangle \left\langle \sigma_l^z(t) \right\rangle,
\end{aligned}
\tag{2.1}
$$

as well as the entanglement entropy between two halves of the system, say $A$ and $\bar{A}$,

$$
S(t) = -\mathrm{Tr}\, \rho_A(t) \log \rho_A(t),
\tag{2.2}
$$

where $\rho_A$ is the reduced density matrix of the subsystem $A$. For the time evolution we used a second order Trotter expansion, with the maximum bond dimension of the matrix product states fixed at 512.

As shown in Figs. 1.1 and 1.2, at a first sight, the time evolutions of the correlation functions $C_z(l, t)$ and the entanglement entropy $S(t)$ look remarkably similar for an anti-confining field, $h < 0$, to the one obtained in the dynamical confinement region, $h > 0$, studied in [31]: in both cases, the light-cone propagation of correlations and the growth of entropy are suppressed, and oscillations are observed.

Closer examination of the simulation results discerns, however, some important differences between the two cases. While correlations have an oscillatory behaviour in both cases, the corresponding *frequencies* and *amplitudes* are quite different. As discussed in Ref. [31], in the confining case $h > 0$ the frequencies of oscillations scale with $h^{2/3}$ and correspond to bound states of domain walls called "mesons". In contrast, for the anti-confining case the characteristic frequency scales with $h$, as shown explicitly by the quench spectroscopy discussed in Subsection 2.4.

### 2.2 Bloch oscillations

Consider first a pure transverse field quench with $h = 0$ and $g > 0$. The post-quench state then consists of independent kink-antikink pairs with momenta $k$ and $-k$. These kinks behave as non-interacting fermions with a dispersion relation, $\epsilon(k) = 2J\sqrt{1 + g^2 - 2g\cos k}$, and propagate with the group velocity

$$
v(k) = \frac{\partial \epsilon(k)}{\partial k},
\tag{2.3}
$$

limited by the maximum velocity, $v_{\max} = 2Jg$. The initial density of kinks can be determined following Refs. [17, 51]. Kink-antikink pairs of momentum $\pm k$ are created with a pair creation amplitude, $K(k) = \tan \Delta_k / 2$, with

$$
\cos \Delta_k = \frac{g g_0 - (g + g_0) \cos k + 1}{\sqrt{1 + g^2 - 2g\cos k}\sqrt{1 + g_0^2 - 2g_0 \cos k}}.
\tag{2.4}
$$

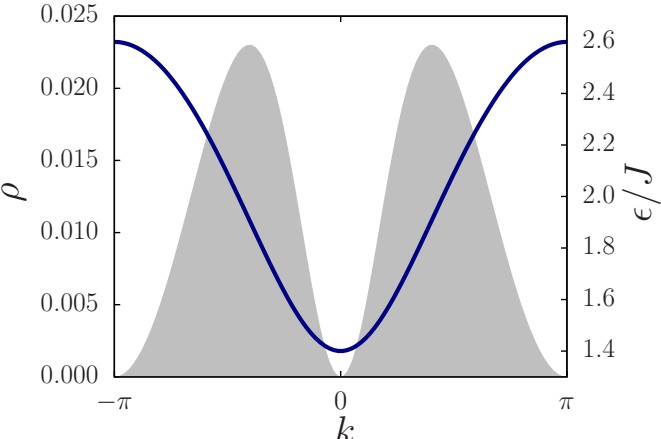

Figure 2.1: The kink dispersion relation $\epsilon(k)$ for $g = 0.3$ (full line) and the initial density $\rho(k)$ (shaded region) for $g_0 = 0$, $g = 0.3$.

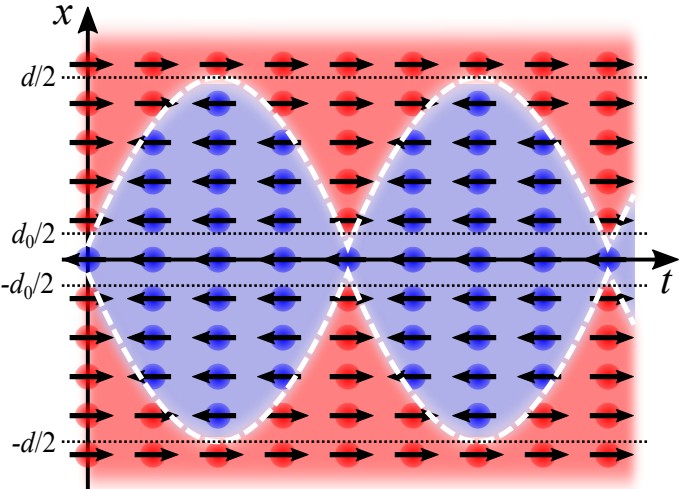

Figure 2.2: Illustration of Bloch oscillation of a bubble with a definite initial momentum of the kinks forming the bubble walls. Arrows indicate dominant spin directions in space and time. Time delays at kink collisions are neglected.

and the density of kink pairs with momentum in the interval $[k, k + dk]$ is given by

$$\rho(k) = \frac{|K(k)|^2}{1 + |K(k)|^2} = \frac{1 - \cos(\Delta_k)}{2} = \sin^2(\Delta_k/2) \,, \tag{2.5}$$

shown in Fig. 2.1. The denominator in this equation simply reflects the fermionic nature of kinks, and the spatial bubble density is just the integral of $\rho(k)$,

$$\rho_{\text{bubble}} = \int_0^\pi \frac{dk}{2\pi} \rho(k) \,. \tag{2.6}$$

Let us consider now a finite longitudinal field $h < 0$. Before entering the issue of bubble dynamics, it is important to discuss the origin of the bubbles. In Coleman's original scenario, the bubbles appear as a result of vacuum tunnelling. The tunnelling probability per lattice site for the Ising spin chain was computed in [52] and in a semi-classical approximation[1] it is

---

[1] We note that the conditions for the semi-classical approximation are that $g$ is sufficiently far away from its

given by

$$\gamma = \frac{\pi g m}{9} \exp\left\{-\frac{1}{mg}|f(-i\log h)|\right\}, \qquad (2.7)$$

where $m = (1-g^2)^{1/8}$ is the spontaneous magnetisation for the pure transverse field Ising spin chain with transverse field $g$ and

$$f(x) = 2\int_0^x \epsilon(k)dk. \qquad (2.8)$$

For the range of parameters considered in our simulations, the nucleation rate per site estimated from (2.7) is very small, not exceeding $10^{-6}$. In fact, the nucleation is heavily suppressed by a mechanism analogous to the Schwinger effect [53] in Quantum Electrodynamics: the creation of the bubbles can also be viewed as spontaneous creation of particle-antiparticle (kink-antikink) pairs in a homogeneous external field (here given by $h$), which is the same as in continuum field theory. However, the bubbles created in the quenches we consider originate from the finite energy density in the initial state which is not fine-tuned to be the false vacuum itself. Nevertheless, the fate of the bubbles after their appearance is essentially independent of the mechanism responsible for their creation and, as we demonstrate, the suppression of their subsequent expansion is due to Bloch oscillations, which arise from the presence of the lattice.

Bloch oscillations can be described using a simple semi-classical picture, similar to the one used in [54] to describe the spectrum of mesonic excitations due to confinement. This is expected to work under the conditions that (i) the system is apart from the immediate vicinity of the critical point $g_c = 1$, (ii) the mean inter-particle spacing is much larger than the correlation length $\xi$: $\rho_{\text{bubble}}\xi \ll 1$, where $\xi$ is of order one away from the vicinity of $g_c$, and (iii) $|h|$ is sufficiently small. The validity of the latter condition can be seen from the fact that the semi-classical description of the meson spectrum (cf. Appendix A) is very accurate for the parameter range considered here [31].

To a first approximation, we can neglect the correction to the dispersion relation $\epsilon(k)$, and treat the dynamics of the bubbles as that of a two-particle system, a kink and an antikink interacting via a repulsive potential

$$V(r) = -\chi\, r, \qquad (2.9)$$

where $r$ is the distance between the kinks, and $\chi$ the coefficient given by the energy gain of flipping the magnetisation into the external field's direction: $\chi = 2m|h| = 2|h|(1-g^2)^{1/8}$. The semi-classical equation of motion of the kinks is then written as

$$\dot{r} = 2\frac{\partial\epsilon(k)}{\partial k}, \qquad \dot{k} = -\frac{1}{2}\frac{\partial V}{\partial r} = \frac{\chi}{2}, \qquad (2.10)$$

with the factors 2 and 1/2 related to having two mobile kinks. The second equation yields immediately $k(t) = k_0 + \frac{1}{2}\chi t$, and can be used to determine the kink-antikink distance as

$$r(t) = \frac{4}{\chi}[\epsilon(k_0 + \chi t/2) - \epsilon(k_0)] + r_0, \qquad (2.11)$$

with $r_0$ initial size of the bubble (typically of the order of the lattice spacing), and $\pm k_0$ the initial momenta of the kinks. Let us first consider bubbles where the initial size $r_0$ can be neglected. Due to the periodicity of $\epsilon(k)$, the kink velocity reverses sign when the momentum

---

critical value $g_c = 1$ and it is valid in the asymptotic limit $h \to 0$, which means that the exponent is large and tunnelling is suppressed.

$k$ passes the boundary of the Brillouin zone at $k = \pm\pi$, where the kinks turn back. As a result, kinks return to their original position and collide again after a period $T(k_0)$ when the bubble re-collapses ($r(T(k_0)) \approx 0$). This happens when $k_0 + \chi T(k_0)/2 = 2\pi - k_0$. At this point, the kink and the antikink are reflected, and start off again with the whole cycle repeating as illustrated in Fig. 2.2, causing the bubbles oscillate in time. The maximum amplitude of these oscillations is obtained when $k_0 = 0$:

$$l_{\max} = \frac{4(\epsilon(\pi) - \epsilon(0))}{\chi} = \frac{8g}{m|h|} \, . \tag{2.12}$$

In contrast to the idealised single-bubble dynamics shown in Fig. 2.2, the observed oscillations shown on the left of Fig. 1.1 result from a large number of bubbles, each having different initial momenta and initial sizes. Nevertheless, these Bloch oscillations still have a characteristic time. As obvious from Fig. 2.1, (and can be indeed verified by direct calculation, for an initial state with $g_0 = 0$) the kink density is highest around momenta corresponding to $v_{\max}$, which determines the front line of the bubbles, and thus the overall frequency of oscillations via the condition, $\frac{1}{2}\chi T_B = 2k_0 \approx \pi$ for small $g$-s, yielding

$$\omega \approx \frac{2\pi}{T_B} = \chi \, .$$

The presence of a distribution of domain wall momenta leads, however, to a gradual decay of oscillations due to the dependence of the oscillation period and phase on the initial kink momentum. Bubble collisions have a similar, degrading effect on coherent oscillations.

We close this subsection with some important observations regarding the role of collisions:

- For small bubbles, kink-antikink pairs collide once during every oscillation period. In principle they could annihilate into mesons then; however our numerics shows no such effect in the accessible time frame, which is consistent with recent findings [55] that the inelastic scattering is very ineffective. In this regard we also note that annihilation into mesons would involve string breaking which is heavily suppressed as can be understood via relating it to the Schwinger effect [38].

  These collisions also give rise to a time delay due to the interaction between kinks. In the case of zero longitudinal field, $h = 0$, however, the kink-antikink scattering amplitude is simply $-1$, and there is no time delay. Therefore any time delay introduced by kink collisions is of order $h$, which we can neglect in the simple semi-classical picture used here.

- Collisions between different bubbles lead to corrections to the simple motion described above. This effect can be neglected if the average spacing between bubbles is larger than their maximum allowed size, which leads to the condition $\rho_{\text{bubble}} \ll 1/l_{\max}$ requiring the longitudinal field to satisfy $h \gg h_c$, where

$$h_c = \frac{8g}{m}\rho_{\text{bubble}} \, , \tag{2.13}$$

  therefore we always use field values larger than $h_c$ in our simulations. The values of $h_c$ are reported in Table 2.1 for different values of $g$.

- For bubbles created with a sufficiently large initial size $r_0 > l_{\max}$, the kinks never collide and instead oscillate around spatially separated positions with frequency given exactly by $\chi$ (cf. also [54]). Note that the creation of large bubbles is suppressed, since creating a finite sized bubble of the true vacuum corresponds to a process involving a number

Table 2.1: Values of $h_c$ for different values of $g$.

| $g$ | 0.20 | 0.25 | 0.30 | 0.35 | 0.40 |
|---|---|---|---|---|---|
| $h_c$ | 0.0041 | 0.0080 | 0.0139 | 0.0223 | 0.0338 |

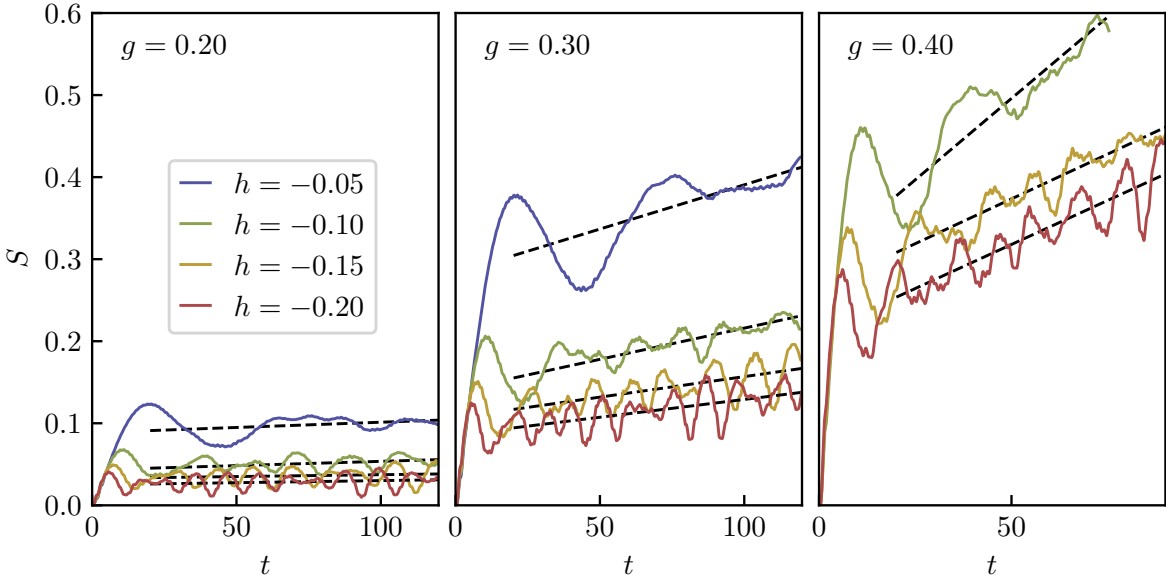

Figure 2.3: Entanglement entropy as a function of time for different values of $g$ and $h$, with the overall drift indicated by dashed lines.

of simultaneous spin flips given by the bubble size $r_0$, with probability suppressed exponentially in $r_0$. As a result, the contribution of 'collisionless bubbles' (those satisfying $r_0 > l_{max}$ and therefore oscillating without periodic internal collisions) increases sharply with decreasing $l_{max}$ i.e. increasing $|h|$, which is consistent with the quench spectroscopy results reported later in Subsection 2.4. In addition, to avoid collisions between kinks from different bubbles, the average bubble spacing must be much larger than $l_{max}$, leading to the same condition $|h| \gg h_c$ as before.

- It is apparent from Fig. 1.2 that even though the growth of entanglement entropy is suppressed, it still shows a slow drift in time in addition to the dominant feature of temporal oscillations. This drift can be understood to originate from bubble collisions, which become less probable as $|h|$ grows compared to $h_c$. Indeed, Fig. 2.3 demonstrates that the drift is suppressed for other values of $g$ as well when $|h| > h_c$. Note that entanglement entropy grows so fast at $g = 0.4$ for $h = -0.05$ that it was not possible to simulate the evolution in the time scale shown in the figure. As can be seen from Table 2.1, $h_c = 0.0338$ for $g = 0.4$, so bubble collisions are much less suppressed than for the other two cases $g = 0.2$ and $0.3$.

## 2.3 Verifying average bubble size and scaling

One piece of evidence for the scenario of Bloch oscillations is that it predicts an average bubble size that agrees reasonably well with the spatial extension of the correlations. This is demonstrated in Figure 1.1, where the blue lines depict the estimate for the average bubble size

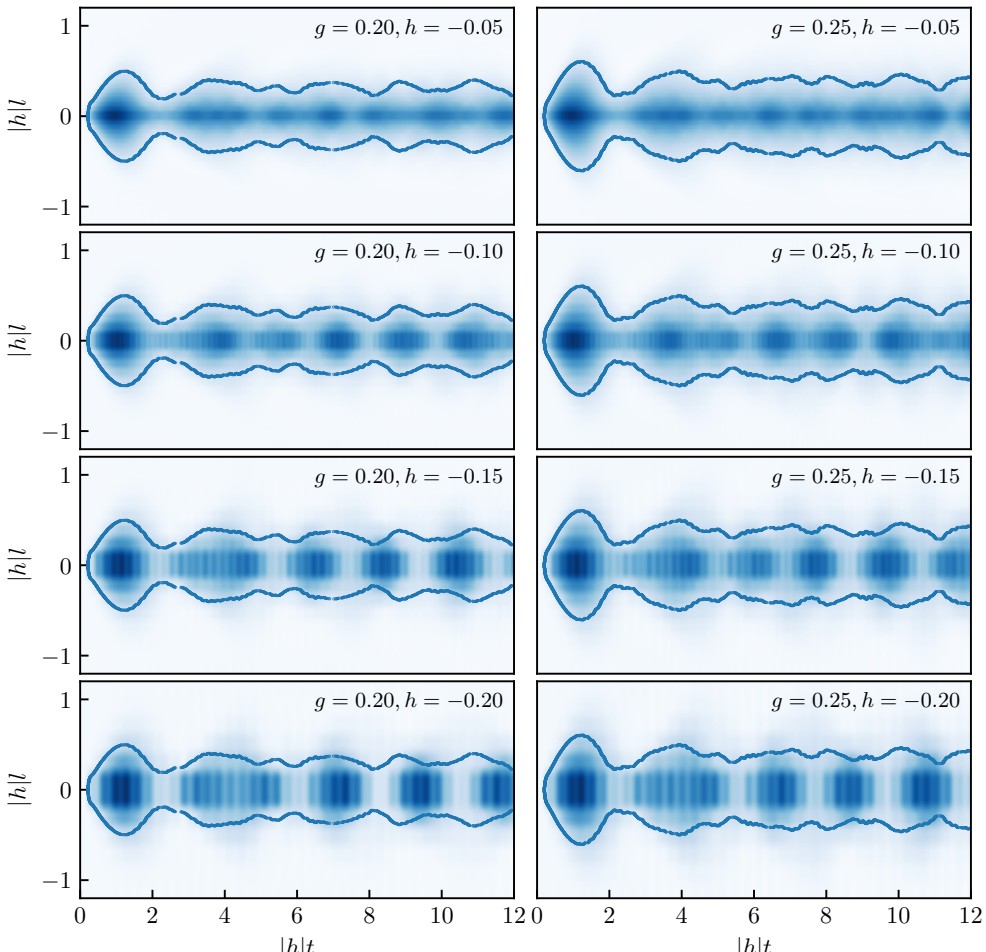

Figure 2.4: Scaling evidence for Bloch oscillations for the transverse field values $g = 0.20$ (left) and $g = 0.25$ (right), with the longitudinal field taking the values $h = -0.05, -0.10, -0.15, -0.20$ for each. The plot shows $\left\langle \sigma_0^z \sigma_l^z \right\rangle_c$ as a function of the scaling variables $|h|t$ and $|h|l$. The colour scale is defined by normalising the maximum value of the correlator to 1, while the blue lines show the bubble wall defined by the data with $h = -0.10$ at one fifth of its maximum value. Note that $h = -0.05$ where the scaling is visibly the least perfect, is a small field for which quench spectroscopy (cf. Subsec. 2.4) reveals that the system is not yet cleanly dominated by Bloch oscillations.

$$\langle r \rangle_{\text{anti-conf}} \approx \frac{1}{\rho_{\text{bubble}}} \int_0^\pi \frac{dk_0}{2\pi} \rho(k_0) \frac{4}{\chi} \left( \epsilon(\pi) - \epsilon(k_0) \right) \tag{2.14}$$

obtained by neglecting the original bubble size $d_0$. For the standard confining case, a similar reasoning gives the average bubble size

$$\langle r \rangle_{\text{conf}} \approx \frac{1}{\rho_{\text{bubble}}} \int_0^\pi \frac{dk_0}{2\pi} \rho(k_0) \frac{4}{\chi} \left( \epsilon(k_0) - \epsilon(0) \right) \tag{2.15}$$

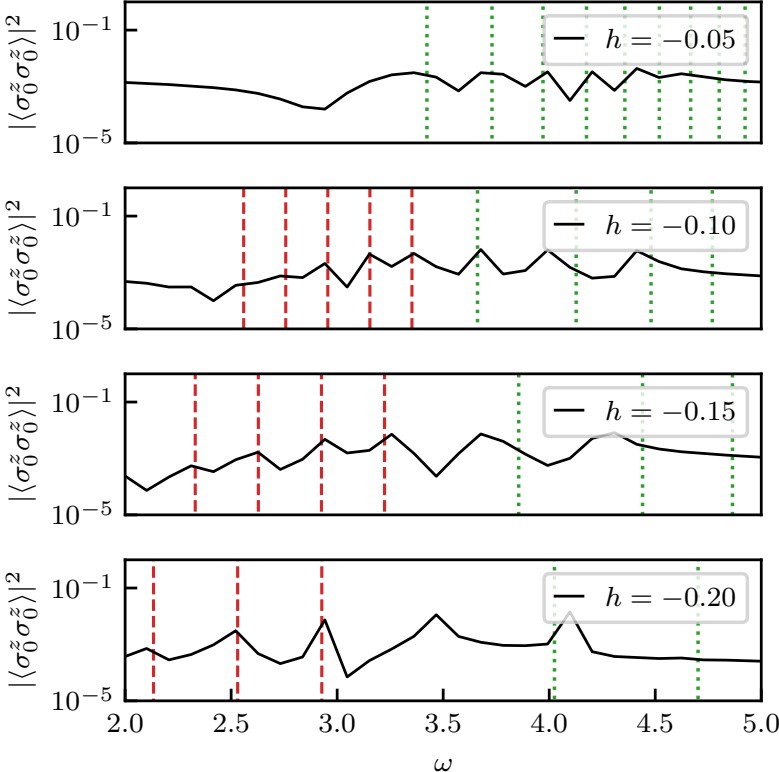

Figure 2.5: Power spectrum of $\left\langle \sigma_0^z(t)\sigma_l^z(t)\right\rangle_c$ for $l = 0$ and $g = 0.25$. Green dotted lines show meson frequencies (computed following [31]) - note that they become less and less relevant as $|h|$ grows. Red dashed lines have a spacing $\chi = 2|h|m$, indicating a regular sequence of higher harmonics of the Bloch oscillations, which instead become more prominent for higher values of $|h|$ as $l_{\max}$ becomes smaller. In the cases shown, the minimum size of collisionless bubbles $l_{\max}$ (from top to bottom) are 40.3, 20.2, 13.4 and 10.1, respectively.

since, in this latter case, kink momenta oscillate between $k_0$ and 0. Note that these estimates depend both on $g$ and $h$, and give a very good estimate for the spatial extension of the correlations shown in Figure 1.1.

Another tell-tale signal of Bloch oscillations is that their spatial extension is predicted to scale as $\sim 1/h$, while their frequency is proportional to $h$. This is manifest in the numerically computed time evolution, as demonstrated in Fig. 2.4, where the space-time bubble contour is extracted from the data at $h = -0.1$ and then superimposed on the time evolution obtained for other $h$ values, with time and space rescaled accordingly. Note that the actual distance and time scales vary by a factor of 4, while the correlation profiles remain almost identical when plotted in terms of the scaled variables, $|h|t$ and $|h|l$. This scaling works remarkably well for the first few oscillations. Differences for larger values of time can be attributed to deviations from our simple semi-classical picture such as the presence of localised spin excitations with frequencies much higher than the Bloch oscillations of the bubble walls, as well as distortions due to bubble collisions and the contributions from the periodic kink collisions when bubbles shrink to their minimal size.

## 2.4   Quench spectroscopy

'Quench spectroscopy', presented in Fig. 2.5, i.e. the Fourier analysis of the time evolution after the quench provides a further tool to assess Bloch oscillations. For negative values of $h$ with small magnitude, we only observe frequency peaks corresponding to mesonic bound states, just as for the case of dynamical confinement with $h > 0$ [31]. The energy gap of these mesonic excitations is always larger than twice the kink gap, grows with increasing $|h|$, and can also be computed theoretically using a semi-classical method [54], which we briefly summarise in Appendix A. For longitudinal fields such that $|h|$ is smaller than, or comparable to $h_c$, we do not observe well-defined frequencies corresponding to Bloch oscillations, as shown by the top plot in Fig. 2.5. This is not unexpected, since in such a case bubble collisions prevent independent Bloch oscillations.

For larger field values $|h| \gg h_c$, collisions between bubbles become less frequent. In addition, the maximum Bloch oscillation length $l_{\max}$ becomes shorter with increasing $h$, and the number of collisionless bubbles increases rapidly. This leads to the appearance of regularly spaced frequency peaks well below the threshold of mesonic excitations, corresponding to higher harmonics of the Bloch frequency $\chi$. Indeed, such a series of peaks is seen to coincide with the red lines in the lower three plots in Fig. 2.5, with the distance between them equal to $\chi$. Note that the simulation can only capture higher harmonics, mostly due to the finite time window of the numerical simulations, but also due to low-frequency background which results from the quench time evolution containing frequencies corresponding to all differences between energy levels of the post-quench Hamiltonian. In addition, the energy needed to create mesonic excitations also increases with $|h|$. We therefore expect, and indeed find, that mesonic contributions to the frequency spectrum become less and less pronounced with growing $|h|$. This stands in stark contrast to the case of dynamical confinement, where meson excitations continue to dominate the time evolution even for large values of the longitudinal field $h$ [31].

# 3   Conclusions

As we demonstrated in this work within the framework of the transverse field Ising model, light-cone time evolution and the decay of the false vacuum can be absent in one-dimensional systems even in a deconfined quench regime, where the formation of bubbles would be energetically favourable. Rather, quite unexpectedly, we observe in this regime spatially confined correlations, oscillating in time, which we identify as Bloch oscillations. These appear due to the underlying lattice and the periodicity of the quasi-particles' dispersion relation in the momentum variable. The absence of relaxation after the quantum quench can also be considered an effect of the bounded quasi-particle dispersion, due to which the domain walls can only carry away a limited portion of the energy that would be liberated by the expansion of the true vacuum bubble.

Our present results demonstrate that Bloch oscillations play a key role in inhibiting the growth of bubbles which result from nucleation of the true vacuum in a global quantum quench starting from a translationally invariant initial state dominated by the false vacuum. The effect we observe can also be interpreted as a suppression of thermalisation in a global quench of a spin chain with (anti-)confining dynamics. Compared to the recent works [34, 38], where Bloch oscillations have been discussed in the context of inhomogeneous states, our choice of initial state allows us to address directly the dynamics of the false vacuum. Also, while the discussion of Refs. [34, 38] relies on perturbation theory in the transverse field $g$, our semi-classical analysis is valid for any $g$ – not too close to the critical point $g = 1$. Comparing the maximum bubble size to the post-quench bubble density yields that the dominance of Bloch

oscillations and the resulting suppression of bubble growth requires a longitudinal magnetic field, $h$, that exceeds a characteristic value $h_c$ (cf. Eq. (2.13)).

Our observations of emergent Bloch oscillations are also in agreement with recent results obtained in kinetically constrained Rydberg spin systems [56]. However, while in Ref. [56] a special constraint (fine-tuning) was needed to generate Bloch oscillations, here they emerge quite generically, without any constraint in a regime where quasi-particle excitations would naively be expected to speed up and spread correlations over the system.

Our results are relevant for experimental realisations of the tunnelling decay of the false vacuum such as put forward in Ref. [22]. We predict that in discrete spin chains, the dynamics after bubble nucleation generally leads to Bloch oscillations, in stark contrast to expectations from continuum quantum field theories. Our results also demonstrate that Bloch oscillations realised in simple spin chains can clearly be identified from the scaling of the space-time dependent spin-spin correlation functions with the applied longitudinal field, and from quench spectroscopy which (for $h \gg h_c$) is dominated by a a series of sharp peaks in the frequency spectrum with a regular spacing, corresponding to collisionless bubbles undergoing Bloch oscillations.

**Note added** - After the submission of this manuscript, the vacuum tunnelling in the Ising spin chain was numerically observed in [57], where the prediction (2.7) was also verified. The observation of the tunnelling was made possible by choosing transverse fields close to the critical value, $g \geq 0.7$ which enhances the transition amplitude (2.7) to magnitudes $10^{-3} - 10^{-1}$, and also by observing the dynamics for times much shorter (at least by an order of magnitude) than the period of Bloch oscillations for the chosen parameter values of $g$ and $h$. It is an interesting issue to perform simulations for parameter values and time ranges where effects of both Bloch oscillations and vacuum tunnelling can be observed, to see how the two mechanisms interfere. However, these regimes are difficult to access by the present numerical methods.

# Acknowledgments

G.T. acknowledges useful discussions with M. Kormos and S. Rutkevich. This work was partially supported by National Research, Development and Innovation Office (NKFIH) through the OTKA Grants FK 132146 and K 138606, the Hungarian Quantum Technology National Excellence Program, project no. 2017-1.2.1- NKP- 2017-00001, and by the Fund TKP2020 IES (Grant No. BME-IE-NAT), under the auspices of the Ministry for Innovation and Technology, and within the Quantum Information National Laboratory of Hungary. M.A.W. has also been supported by the ÚNKP-21-4 New National Excellence Program of the Ministry for Innovation and Technology from the source of the National Research, Development and Innovation Fund.

# A  Semi-classical meson spectrum

Here we summarise briefly the semi-classical approach to the meson spectrum developed in [54]. Consider a kink and an antikink moving under a linear confining potential with the Hamiltonian

$$H = \epsilon(k_1) + \epsilon(k_2) + \chi |x_1 - x_2| . \tag{A.1}$$

Introducing variables describing the centre-of-mass and relative motion

$$X = \frac{x_1 + x_2}{2} \quad , \quad x = x_1 - x_2$$
$$K = k_1 + k_2 \quad , \quad k = \frac{k_1 - k_2}{2} \tag{A.2}$$

the Hamiltonian takes the form

$$H = \omega(k; K) + \chi |x| \quad , \quad \omega(k; K) = \epsilon(k + K/2) + \epsilon(k - K/2). \tag{A.3}$$

For the relative motion, swapping the role of the canonical coordinates by introducing $p = -x$ as momentum and $q = k$ as coordinate, the Hamiltonian describes the motion of a particle with "kinetic energy" $\chi |p|$ in a one-dimensional "potential" $\omega(q; K)$. The simplest way to obtain the energy levels is to use Bohr-Sommerfeld quantisation. For $K < 2 \arccos g$ the potential $\omega(q; K)$ has a single minimum at $q = 0$, and the quantisation condition reads

$$2E_n(K) q_a(n, K) - \int_{-q_a(n,K)}^{q_a(n,K)} dq\, \omega(q; K) = 2\pi \chi (n - 1/4), \tag{A.4}$$

with $n = 1, 2, \ldots$, and the turning point $q_a(n, K) \in [0, \pi]$ are determined by the equation

$$\omega(q_a(n, K); K) = E_n(K). \tag{A.5}$$

For $K > 2 \arccos g$ the potential $\omega(q; K)$ has two minima. For $E > \omega(0, K)$ the above treatment is unchanged; however, for $E < \omega(0, K)$ the motion takes place in one of the two separated potential wells and the quantisation condition changes to

$$E_n(K)[q_a(n, K) - q_b(n, K)] - \int_{q_b(n,K)}^{q_a(n,K)} dq\, \omega(q; K) = \pi \chi (n - 1/2), \tag{A.6}$$

with $n = 1, 2, \ldots$, and the turning points $q_{a,b}(n, K) \in [0, \pi]$ determined by

$$\omega(q_{a,b}(n, K); K) = E_n(K). \tag{A.7}$$

In principle, the Bohr-Sommerfeld quantisation gives the leading asymptotic behaviour of the energy for large quantum numbers $n$; however, for the values of $g$ and $h$ considered here it is known to give accurate values for the meson masses even for small $n$ [31], and so there is no need to resort to the more sophisticated methods detailed in [54].

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
