# Peer review of "Bloch oscillations and the lack of the decay of the false vacuum in a one-dimensional quantum spin chain"

_SciPost Physics, doi:SciPost Phys. 12, 061 (2022)_

## Round 2 · Referee Report · Anonymous (Referee 1) · 2021-10-20

Strengths

  • Broad appeal
  • Very accessible and easily readable

Weaknesses

  • The results lack novelty to some extent

Report

The authors study the spreading of correlation and entanglement in quantum quenches in the one-dimensional Ising spin chain, in the presence of both a transverse and longitudinal field. When considered in a pure transverse field, the model is equivalent to free fermions and thus easily solvable, but a weak longitudinal field dramatically changes the setup. Indeed, excitations over the ferromagnetic phase (i.e. domain walls) experience a pairwise linear potential proportional to the longitudinal field and hence experience confinement. In particular, the authors closely follow the investigation of [Nat. Phys. 13 (2017) 246-249], but with a longitudinal field of opposite sign. In this case, the system initially lies in the false vacuum of the confining theory and the quench excites bubbles of true vacuum in the false-vacuum sea. In this setup, domain walls feel a repulsive force and, in the case of a continuum theory, this would cause the domain walls to indefinitely drift far apart. On the lattice, the finiteness of the kinetic energy changes this picture and the domain walls undergo Bloch oscillations, resulting in false vacuum bubbles with a finite maximum extent.

The study of confinement in condensed matter setups attracted a lot of interest in recent times, therefore I believe this work is timely and of broad appeal. However, I have a main criticism concerning the originality of the results here presented. The authors often use adjectives like "quite counter-intuitively" and "unexpectedly" while referring to bubbles of finite length despite the presence of a repulsive interaction, but the crucial role of the lattice in inducing Bloch oscillations has already been pointed out in Refs. [34] and [38] of the manuscript. In Ref. [34] a domain wall in the confining Ising has been studied: in a continuum theory, the domain wall would accelerate towards the false-vacuum region, but due to lattice-induced Bloch oscillations the domain wall oscillates around its original position. Then, in Ref. [38] suppression of transport is studied in the confining Ising in the case of dilute domain walls. These domain walls are essentially experiencing Stark localization in a linear potential induced by the surrounding excitations, hence they undergo Bloch oscillations around their original position. Of course, this effect would be absent in a continuum theory. The setup studied by the authors is different from that of the above references, nevertheless the general mechanism inducing finite-sized vacuum bubbles has already been presented there: I invite the authors to give proper credit to Refs. [34] and [38] and reconsider the choice of wording concerning the novelty of the phenomenon. For this reason, I am not sure if the acceptance criteria of SciPost Physics are met, but I support publication in SciPost Core after my detailed comments have been properly addressed.

Detailed comments:

  • pg 2, "The resulting confining force [41] inhibits the propagation of the domain walls to large distances, and prevents thermalization of the system within all time scales accessible to numerical simulations."

I think Ref. [38] should be properly mentioned here in connection with the absence of the Schwinger mechanism on very long time scales.

  • Fig. 1.2. The (approximately linear) entanglement growth signals the presence of excitations of finite velocity in the system. Two possible effects come to my mind. The first one is the Schwinger effect: new mesons are dynamically produced due to string breaking. The second mechanism is: mesons are too dense and they collide during the expansion, causing scattering and becoming traveling excitations. Can the authors comment on which of the two mechanisms (or maybe another one I could not think of) is responsible for the drift? I think that by reducing (in modulus) the longitudinal field, the meson becomes shallower (i.e. it favors the second mechanism) and string breaking becomes harder (i.e. it mitigates the first effect). Checking whether the linear growth increases or diminishes after this operation should help in characterizing the underlying mechanism.

  • pg 7. "Kink-antikink pairs collide once during every oscillation period. They could, in principle, annihilate into mesons then; however our numerics shows no such effect in the accessible time frame, which is consistent with recent findings [52] that the inelastic scattering is very ineffective"

I believe Ref. [38] should be mentioned here, in particular the result that the Schwinger effect takes place on exponentially long time scales.

  • pg 7. "This requires the longitudinal field to be larger than some minimum value, which can be computed as a function of g and turns out to be smaller than the fields used in our simulations."

I think it could be worth giving the typical value of $2\epsilon(\pi)\rho_{bubble }\chi $

  • pg 7. "The energy gap of these mesonic excitations is always larger than twice the kink gap, grows with increasing |h|, and can also be computed theoretically using a semi-classical method [53, 31]. For small quench fields, we do not observe frequencies corresponding to Bloch oscillations"

I am familiar with the method, but I think a short summary should be provided here as well. This will make the paper more self-contained and easily accessible for a broader audience.

  • pg 7 (and below) "For small quench fields, we do not observe frequencies corresponding to Bloch oscillations." and the following discussion.

I think the authors should be more specific in this discussion and use better wording. Indeed, in the limit of a weak transverse field the size of the bubble is governed by the ratio h/g. If h/g is large, the bubble is small. Maybe the authors should be more specific rather than writing "small quench field", since one can still have g and h both small, but h/g large (thus small bubbles). I invite the authors to clarify the quoted sentence and the following discussion.

---

## Round 2 · Referee Report · Anonymous (Referee 2) · 2021-10-25

Strengths

1) The manuscript address a timely and challenging topic which is the suppression of the false vacuum decay in many-body quantum systems displaying confinement of excitations.

2) The numerical analysis presented is detailed.

3) The manuscript is clearly written and it can be understood also by non-experts in the field.

Weaknesses

1) A discussion and a quantification of the timescales required for the decay of the false vacuum is missing (see also the report).

Report

The manuscript addresses the problem of the suppression of the false vacuum decay in the Ising model with both transverse $h$ and longitudinal $g$ field, which is a paradigmatic model displaying confinement of the quasi-particle excitations. The false vacuum decay consists in the transition of an unstable-false vacuum state to the stable-true vacuum by means of the nucleation of bubbles of the true vacuum inside the false one. This problem is timely and challenging as the false vacuum decay is a non-perturbative effect which is very hard to treat analytically and to access numerically due to the large timescales involved.

Within this framework, the analysis of the paper addresses the suppression of the false vacuum decay in quantum quenches from the false vacuum (ground state for $h>0$) to the anti-confining ($h<0$) regime of the quantum Ising chain. In particular, the false vacuum is observed not to decay for the time scales the authors numerically access. The system, instead, displays oscillations in the (suppressed) entanglement growth and in the (suppressed) correlation spreading . The oscillatory motion is ascribed to Bloch oscillations. These statements are supported by a detailed numerical analysis based on the iTEBD method. The analysis presented, in particular, clearly distinguishes the different physics of the oscillations happening in the confining regime ($h>0$-meson spectrum) from the one of the Bloch oscillations happening in the anti-confining regime ($h<0$ with frequency of oscillations $\propto h$) .

Based on these facts, I think the manuscript deserves publication in Scipost Physics. However, some aspects of the discussion must be improved . Once the authors thoroughly address the Requested changes below, I will be happy to recommend the manuscript for publication in Scipost Physics.

Requested changes

1) The authors do not discuss at all the relevant timescales of the false vacuum decay. I am aware of the fact that this is a very difficult problem and I do not demand the authors to find a formula for this aspect. However, the authors should mention and discuss the result of Phys. Rev. B 60, 14525, which gives an analytical prediction for the timescale $\tau$ of the decay of the false vacuum because of bubbles nucleation in the Ising chain as for $h \to 0^{-}$ and $g$ not too close to $1$. This is, to my knowledge, the only theoretical prediction for this challenging problem and it is therefore extremely relevant for the analysis of the paper. I therefore expect the authors to discuss the relation between such a result and the outcome of their numerical simulations. I would expect that the value of $\tau$ is far too large to be able to observe numerically the false vacuum decay for the values of the post-quench parameters $(g,h)$ used by the authors (when the formula for $\tau$ applies).

2) In the abstract and in the introduction-conclusions the claim is made that "the suppressed decay is a lattice effect". I find this claim in general not correct since the suppression of the false vacuum decay is present also in the continuum/field theory description. The mechanism underlying the decay of the false vacuum by bubbles formation is analogous to the Schwinger mechanism of QED [Phys. Rev. 82 (1951) 664], where particle/anti-particle pairs are produced out of the false vacuum from a quench of the background electric field. As a matter of fact, the decay rate of the false vacuum predicted by the Schwinger effect presents an exponential form similar to the one computed in the aforementioned Reference Phys. Rev. B 60, 14525. As a consequence I would not classify the suppression of the false vacuum decay as a lattice effect since it can be suppressed also in field theory. For the specific case of the lattice, the mechanism underlying the suppression of the false vacuum decay is given by the Bloch oscillations, but this does not imply that the same process cannot happen also in the continuum. The authors should therefore change the discussion in the abstract and in the other parts of the manuscript, such as the conclusions, accordingly.

3)The prediction of the authors for the spatial extention of the vacuum bubbles is based on the semiclassical two-fermion method, first proposed for the Ising chain in [Journal of Statistical Physics volume 131, pages917–939 (2008)]. The authors should cite this reference and accordingly discuss the physical meaning and the regime of parameters $(g,h)$ where this approximation is expected to work. This aspect must be clearly discussed before Eq.(2.7)

4) At the beginning of page 7 the authors say that "kink-antikink pairs collide once during every oscillation period". I think this is not true in general. If the initial length $r_0$ of the bubble is larger than the amplitude of the oscillations of the kink-antikink (set by $h$ and $g$) at the extremes of the bubble, then the extremes just perform independent Bloch oscillations.

Some minor points 5) On page 2 the authors should specify the boundary conditions used for the quantum Ising chain.

6) Regarding Eq.(2.2) for the entanglement entropy, the authors should specify where the bipartition is done.

7) Regarding the iTEBD simulations, the authors should mention the maximum bond dimension used in the simulations.

---

## Round 3 · Referee Report · Anonymous (Referee 1) · 2021-12-5

Strengths

Clear and easy to read

Weaknesses

The results are based on the interplay of confinement and Bloch oscillations, which has already been investigated in the literature to some extent

Report

I have read the resubmitted version of the manuscript, I believe that all my concerns (as well as those raised by the other referee's report) have been thoroughly addressed.
I think several important points have been now clarified and properly discussed and the manuscript greatly improved.
However, while this investigation is certainly of interest for a large community and adds insight into the physics of confinement, my main concern on the novelty of the results remains unresolved.
As pointed out in my previous report, the interplay between confinement and lattice-induced Bloch oscillations is the core of the investigation of [PRB 99 (18), 180302; PRB 102 (4), 041118]. I agree that the setup is different (in the quoted references, the domain walls are so far apart to undergo independent Bloch oscillations, while in the case studied by the authors the domain walls within a meson are much closer and scatter), but I believe most of the story is already there.
In addition, the methods used to investigate the setup (single-meson semiclassical quantization, quench spectroscopy) are essentially those presented in [Nat. Phys. 13 (3), 246-249].
Hence, while I agree that the framework investigated by the authors is surely a natural and interesting scenario that was not addressed so far, in my opinion the manuscript does not meet the Scipost Physics criteria in terms of novelty. I think the reader with some expertise on the topic would not find the effect "surprising".
Therefore, I look at Scipost Physics Core as a more appropriate venue to convey the authors' findings.
  • validity: high
  • significance: high
  • originality: good
  • clarity: top
  • formatting: perfect
  • grammar: perfect

Author:  Gabor Takacs  on 2021-12-05  [id 2009]

(in reply to Report 1 on 2021-12-05)
Category:
reply to objection

First of all, I am happy that the referee finds that the manuscript has greatly improved, and that it is of interest for a large community, adding insight to the physics of confinement.

I still maintain that Scipost Physics is the appropriate venue to publish our findings. Below I address the concerns raised about the novelty of the results.

Naturally, nothing is surprising when working backwards from a known explanation: take e.g. the findings of the paper the referee mentioned [Nat. Phys. 13 (3), 246-249] in the light of the confinement mechanism. Note that confinement also occurs in quantum field theory, and there is not much difference from the lattice case, except that some of the diagnostic quantities (such as time evolution of entanglement entropy) are much harder to access computationally in the continuum QFT.

However, changing the sign of the force from attractive to repulsive a completely different dynamics is expected, which is well-known in quantum field theory as the decay of the false vacuum. Therefore, finding no qualitative difference in the evolution of the correlations and entanglement entropy on the spin chain compared to the confining case is striking. Even accepting that lattice dynamics is different, looking at our Fig. 1.1 one wonders how the two cases can appear so similar! I do not think that this easily or obviously follows from previous works, including those mentioned by the referee.

Note that even realising the existence of Bloch oscillations in the system, it is not so straightforward to conclude that they really explain the effect. As we point out, it is important that the frequency of bubble collisions depends on the parameters (which is now also quantitatively emphasized by Table 2.3, and is also underlined by the dependence of entropy generation on the parameters, now added as Fig. 2.3), and it is also necessary to factor in the recent result that particle production is suppressed in kink collisions [arXiv:2012.07243, Ref. 55 in the paper]. Quench spectroscopy (even though it was not "invented" in this work) is another decisive piece of evidence that Bloch oscillations indeed play a major role in the time evolution of the system. Also note the additional supporting evidence provided by the scaling of the profiles in the variables |h|t and |h|l.

Finally, I also note that while many of the methods overlap with the work [Nat. Phys. 13 (3), 246-249], the goal of this paper was not developing new methods, but to answer the question which can simply be put as follows: why does the time evolution looks so similar to the dynamical confinement scenario when changing the sign of the longitudinal field?

---

## Round 3 · Referee Report · Anonymous (Referee 2) · 2021-12-27

Strengths

1) The manuscript addresses a timely, interesting and challenging topic in non-equilibrium many-body physics.

2) The analysis is not too technical and therefore the manuscript can be read by a big readership of physicists.

Weaknesses

1) The relation between the results of the manuscript and some previous results of the literature is not thoroughly addressed.

Report

I read the revised version of the manuscript and I really appreciated the efforts of the authors to clarify the unclear points raised in my report (and in the report of the other Referee). I am, in particular, fully satisfied by the discussion of the timescale of the false vacuum decay around Eq.(2.7) and by the discussion of the validity of the semi-classical approximation before Eq.(2.10) . The discussion on page 8, together with Fig.2.3, is also useful to understand the role of bubble collisions in the non-equilibrium dynamics (in particular for the entanglement entropy).

I think that the presentation is greatly improved so that the manuscript can be now read in a self-contained way by a large community of physicists interested in the broad field of non-equilibrium quantum systems.

There is a point (mentioned also by the other Referee) that needs to be addressed properly:

1) In the conclusion section the authors added a paragraph where they discuss the relation between the present work and Refs.[34] and [38] ("Besides the difference from our case [...]"). I think that this part must be improved. Indeed, also in the aforementioned references the effect of the Bloch oscillations can be clearly distinguished from the dynamical confinement regime depending on the values chosen for the parameters $h$ and $g$.

In particular, the analysis of these works is perturbative in the transverse field $g$, and therefore it applies for small values of $g$ and arbitrary values of the longitudinal field $h$. The maximal length the bubbles reach in the Bloch oscillation is therefore $l_{\mathrm{max}}=g/h$.

The analysis of the present manuscript, instead, relies on the semi-classical approximation of [J. Stat. Phys. 131 (2008) 917–939], that applies for small $h$ and $g$ not to close to 1 (in units of $J$). The maximal amplitude of the oscillations is therefore given by Eq.(2.12) and Bloch oscillations are dominant only for $h>h_c$, with $h_c$ in Eq.(2.13)

This is, in my view, the most important difference at the technical level between this manuscript and the aforementioned Refs.[34] and [38] . From the conceptual point of view, however, there isn't any big difference apart from the class of initial states considered. In both the cases the ensuing physics is, as a matter of fact, the one of Bloch oscillations, with the consequent lack of the false vacuum decay, persistent oscillations and absence of thermalization within the numerically accessible time scales.

This is, in my view, the proper way to address and discuss the relation of the present work with the pre-existing literature on the subject (confinement and Bloch oscillations). For this reason I think that the discussion in the conclusion section has to be improved since the present is a bit too superficial. For the same reason, I think that the relation with the aforementioned references must be more deeply discussed also in the introduction section. In the current version there is just a quite generic sentence ("direct observation of Bloch oscillations [...]").

Based on this discussion, the reply the authors provided on this point (in the revised version of the manuscript and in the online form) seems to me tangential and not complete.

The manuscript can be accepted in Scipost Physics once the authors thoroughly address the previous point.

---

## Round 3 · Author Response

We thank the referees for their careful reading of the manuscript and for the constructive criticism which helped us to improve the presentation of the results. We list our replies to their comments below, organised by the points raised in their reports.

---

## Round 3 · List of Changes

Replies to Report 1

1) The authors do not discuss at all the relevant timescales of the false vacuum decay. I am aware of the fact that this is a very difficult problem and I do not demand the authors to find a formula for this aspect. However, the authors should mention and discuss the result of Phys. Rev. B 60, 14525, which gives an analytical prediction for the timescale τ of the decay of the false vacuum because of bubbles nucleation in the Ising chain as for h→0− and g not too close to 1. This is, to my knowledge, the only theoretical prediction for this challenging problem and it is therefore extremely relevant for the analysis of the paper. I therefore expect the authors to discuss the relation between such a result and the outcome of their numerical simulations. I would expect that the value of τ is far too large to be able to observe numerically the false vacuum decay for the values of the post-quench parameters (g, h) used by the authors (when the formula for τ applies).

Reply: We included the reference and the estimates after (2.6). The nucleation of the critical bubble by vacuum tunneling does takes a very long time (per lattice site) for the cases we study. However, our quenches do not start sitting exactly in the false vacuum (before the quench, g=0) so the bubbles are mostly nucleated from the additional finite energy density pumped in by the quench. The main observation of the paper concerns the expansion of the nucleated bubbles, not the vacuum tunneling proper.

2) In the abstract and in the introduction-conclusions the claim is made that "the suppressed decay is a lattice effect". I find this claim in general not correct since the suppression of the false vacuum decay is present also in the continuum/field theory description. The mechanism underlying the decay of the false vacuum by bubbles formation is analogous to the Schwinger mechanism of QED [Phys. Rev. 82 (1951) 664], where particle/anti-particle pairs are produced out of the false vacuum from a quench of the background electric field. As a matter of fact, the decay rate of the false vacuum predicted by the Schwinger effect presents an exponential form similar to the one computed in the aforementioned Reference Phys. Rev. B 60, 14525. As a consequence I would not classify the suppression of the false vacuum decay as a lattice effect since it can be suppressed also in field theory. For the specific case of the lattice, the mechanism underlying the suppression of the false vacuum decay is given by the Bloch oscillations, but this does not imply that the same process cannot happen also in the continuum. The authors should therefore change the discussion in the abstract and in the other parts of the manuscript, such as the conclusions, accordingly.

Reply: It is the suppression of bubble expansion is an effect that is specific to the lattice. The suppression of vacuum tunneling is by a different mechanism, and is related to the size of instanton action determining the tunneling (which is identical to the Schwinger effect in one spatial dimension). We added a sentence to the abstract to clarifying this point, and added a more detailed discussion after eqn. (2.8).

3) The prediction of the authors for the spatial extension of the vacuum bubbles is based on the semiclassical two-fermion method, first proposed for the Ising chain in [Journal of Statistical Physics volume 131, pages 917–939 (2008)]. The authors should cite this reference and accordingly discuss the physical meaning and the regime of parameters (g,h) where this approximation is expected to work. This aspect must be clearly discussed before Eq. (2.7).

Reply: We included the reference, it is before eqn. (2.9) in the new version. Its validity is also discussed in the extended text between eqn. (2.9).

4) At the beginning of page 7 the authors say that "kink-antikink pairs collide once during every oscillation period". I think this is not true in general. If the initial length r0 of the bubble is larger than the amplitude of the oscillations of the kink-antikink (set by h and g) at the extremes of the bubble, then the extremes just perform independent Bloch oscillations.

Reply: Indeed, we were sloppy in our discussion here, and extended the discussion to clarify this point. Such a scenario requires that the kinks start far from each other in space, which means that we should nucleate a large bubble – which is suppressed by bubble size. In fact, the regularly spaced harmonics observed in quench spectroscopy are expected to result from such “collisionless” bubbles, and the dependence of the appearance of these harmonics on the value of |h| is entirely consistent with the expectations. We added new discussion at the end of Subsection 2.2, and also in Subsection 2.4.

5) On page 2 the authors should specify the boundary conditions used for the quantum Ising chain.

Reply: We added a sentence after eqn. (1.1) which specifies that we consider the model in infinite volume, which is also where the numeric approach iTEBD works. Therefore, boundary conditions are not relevant in our context.

6) Regarding Eq.(2.2) for the entanglement entropy, the authors should specify where the bipartition is done.

Reply: Since the volume is infinite (cf. the clarification we made in response to 5) above), the point of bipartition is not relevant, and is indeed not specified in the iTEBD calculation either.

7) Regarding the iTEBD simulations, the authors should mention the maximum bond dimension used in the simulations.

Reply: This information was added after eqn. (2.2).

Replies to Report 2

The study of confinement in condensed matter setups attracted a lot of interest in recent times, therefore I believe this work is timely and of broad appeal. However, I have a main criticism concerning the originality of the results here presented. The authors often use adjectives like "quite counter-intuitively" and "unexpectedly" while referring to bubbles of finite length despite the presence of a repulsive interaction, but the crucial role of the lattice in inducing Bloch oscillations has already been pointed out in Refs. [34] and [38] of the manuscript. In Ref. [34] a domain wall in the confining Ising has been studied: in a continuum theory, the domain wall would accelerate towards the false-vacuum region, but due to lattice-induced Bloch oscillations the domain wall oscillates around its original position. Then, in Ref. [38] suppression of transport is studied in the confining Ising in the case of dilute domain walls. These domain walls are essentially experiencing Stark localization in a linear potential induced by the surrounding excitations, hence they undergo Bloch oscillations around their original position. Of course, this effect would be absent in a continuum theory. The setup studied by the authors is different from that of the above references, nevertheless the general mechanism inducing finite-sized vacuum bubbles has already been presented there: I invite the authors to give proper credit to Refs. [34] and [38] and reconsider the choice of wording concerning the novelty of the phenomenon.

Reply: We included an extended discussion in the second paragraph of the conclusions. It is true that Bloch oscillations are well-known (they are eventually textbook material in condensed matter). However, we still consider the effect surprising as the system does not show the thermalisation dynamics that one would naively expect by an intuition guided by quantum field theory and especially the false vacuum decay scenario. It is also nontrivial why the Bloch oscillations are so effective in suppressing bubble expansion (and therefore thermalisation), since there are other relevant processes such as bubble collisions. Note that the system is in an anti-confining situation, and still the (non)-spreading of correlations observed in the iTEBD simulation looks very similar to the case of dynamical confinement. To explain this, a careful consideration of the relation between post-quench bubble density and Bloch oscillation length is needed (c.f. the updated discussion including the characteristic field strength hc).

Detailed comments: - pg 2, "The resulting confining force [41] inhibits the propagation of the domain walls to large distances, and prevents thermalization of the system within all time scales accessible to numerical simulations." I think Ref. [38] should be properly mentioned here in connection with the absence of the Schwinger mechanism on very long time scales.

Reply: We included a sentence in the introduction. The relevance of the Schwinger effect (which is the same as Coleman’s original instanton picture for tunneling) is mentioned later in the text), and we discuss references [34] and [38] in more detail in the conclusions.

  • Fig. 1.2. The (approximately linear) entanglement growth signals the presence of excitations of finite velocity in the system. Two possible effects come to my mind. The first one is the Schwinger effect: new mesons are dynamically produced due to string breaking. The second mechanism is: mesons are too dense and they collide during the expansion, causing scattering and becoming traveling excitations. Can the authors comment on which of the two mechanisms (or maybe another one I could not think of) is responsible for the drift? I think that by reducing (in modulus) the longitudinal field, the meson becomes shallower (i.e. it favors the second mechanism) and string breaking becomes harder (i.e. it mitigates the first effect). Checking whether the linear growth increases or diminishes after this operation should help in characterizing the underlying mechanism.

Reply: We added figure 2.3 and extended our discussion at the end of Subsection 2.2 to clarify this point.

  • pg 7. "Kink-antikink pairs collide once during every oscillation period. They could, in principle, annihilate into mesons then; however our numerics shows no such effect in the accessible time frame, which is consistent with recent findings [52] that the inelastic scattering is very ineffective" I believe Ref. [38] should be mentioned here, in particular the result that the Schwinger effect takes place on exponentially long time scales.

Reply: We included this statement in the text.

  • pg 7. "This requires the longitudinal field to be larger than some minimum value, which can be computed as a function of g and turns out to be smaller than the fields used in our simulations." I think it could be worth giving the typical value of 2ϵ(π)ρbubbleχ

Reply: We put in a more detailed discussion and added Table 2.1 showing the typical values of the minimum field as a function of g.

  • pg 7. "The energy gap of these mesonic excitations is always larger than twice the kink gap, grows with increasing |h|, and can also be computed theoretically using a semi-classical method [53, 31]. For small quench fields, we do not observe frequencies corresponding to Bloch oscillations" I am familiar with the method, but I think a short summary should be provided here as well. This will make the paper more self-contained and easily accessible for a broader audience.

Reply: Added as Appendix A.

  • pg 7 (and below) "For small quench fields, we do not observe frequencies corresponding to Bloch oscillations." and the following discussion. I think the authors should be more specific in this discussion and use better wording. Indeed, in the limit of a weak transverse field the size of the bubble is governed by the ratio h/g. If h/g is large, the bubble is small. Maybe the authors should be more specific rather than writing "small quench field", since one can still have g and h both small, but h/g large (thus small bubbles). I invite the authors to clarify the quoted sentence and the following discussion.

Reply: We made this discussion more precise now, extending the text of Subsection 2.3 and also added relevant details to the caption of Fig. 2.5.

---

## Round 4 · Author Response

We added text to the introduction, and also a detailed discussion to the conclusion, regarding the results of Refs. [34] and [38].

---

## Editorial Decision

published